# Genotype and growth rate influence female mate preference in *Xiphophorus multilineatus*: Potential selection to optimize mortality-growth rate tradeoff

Meredith Fitschen-Brown[1,2][�]*, Molly Morris[1,2][�]

**1** Department of Biological Sciences, Ohio University, Athens, OH, United States of America, **2** The Ohio Center for Ecological and Evolutionary Studies, Ohio University, Athens OH, United States of America

she These authors contributed equally to this work.

* mf854617@ohio.edu

**Data Availability Statement:** All relevant data are within the paper and its Supporting Information files.

## Abstract

The extent to which mate preferences are adaptive requires a better understanding of the factors that influence variation in mate preferences. *Xiphophorus multilineatus* is a live-bearing fish with males that exhibit alternative reproductive tactics (courter/sneaker). We examined the influence of a female's genotype (courter vs sneaker lineage), growth rate, and social experience on mate preference for courter as compared to sneaker males. We found that females with a sneaker genotype and slower growth rates had stronger mate preferences for the faster growing courter males than females with a courter genotype, regardless of mating experience with one or both types of males. In addition, the relationship between strength of preference and growth rate depended on a females' genotype; females with sneaker genotypes decreased their preference as their growth rates increased, a pattern that trended in the opposite direction for females from the courter genotypes. Disassortative mating preferences are predicted to evolve when heterozygous offspring benefit from increased fitness. Given male tactical dimorphism in growth rates and a mortality-growth rate tradeoff previously detected in this species, the variation in mating preferences for the male tactics we detected may be under selection to optimize the mortality-growth rate tradeoff for offspring.

## Introduction

Mate preference is an important evolutionary process that drives the exaggeration of phenotypic characters [1], influences the trajectories of population size, and accelerates speciation/extinction events [2,3]. Variation in female mate preferences has been suggested to dampen selection on individual male traits and maintain genetic variation [4] which can lead to maintaining multiple male traits or genetic polymorphisms [5]. Examining the different factors that create variation in female mate preferences provides insights into not only when and how mate preference will vary, but also the costs and benefits that drive the evolution of mate

**Funding:** This project was supported in part by the Ohio Center of Ecological and Evolutionary Studies graduate research fellowship program and a Baker Grant from Ohio University. The funders had no role in study design, data collection and analysis, decision to publish, or preparation of the manuscript.

**Competing interests:** The authors have declared that no competing interests exist.

preferences. Teasing apart the different factors that influence mate preferences may require studies that consider how the environment, a female's physiology, and genetics all interact to produce variation in mate preference.

There is considerable evidence that both external and internal factors can alter female mate preferences [reviewed by 6,7]. For example, social interactions with available mates is an external factor that can also produce internal influences on mate preferences when females mate. Virgin females might be less choosy compared to experienced females because virgin females have not yet secured fertilization of the eggs [8], and depending on the available choices and a female's mate choice strategy, fewer available males could alter mate preferences as the plasticity continues after mating [9–13]. And finally, genetic variation in mate preferences is a prerequisite for mate preferences to evolve [14–16] and yet we understand little about how female genotype influences mate preferences [17–19]. This is complicated by the fact that the heritable aspects of variation in preference can remain cryptic if genotypes are responding differently to environmental change [20].

We examined factors influencing variation in the strength of mate preference in a system where previous work provides us with extensive knowledge about the males that females choose and the factors influencing female mate preferences. The high-backed pygmy swordtail fish, *Xiphophorus multilineatus* is a freshwater live-bearing fish from the Pánuco River Basin in Mexico [21]. Males exhibit two alternative reproductive tactics (ARTs). Courter males are larger, deeper bodied [22], and faster growing [23] while reaching sexual maturity later. Sneaker males are smaller, more narrow bodied, and slower growing, reaching sexual maturity earlier [24,25]. The courter males are fixed in their use of courtship behavior, while the sneaker males are plastic depending on social context, using both sneak-chase and courtship mating behaviors (24). Females on average prefer to mate with courter males compared to sneaker males [26]. However, there is variation in this preference depending on both social and sexual experience [27], and variation in this preference may function to help maintain the two ARTs [26].

Variation in male size and age at sexual maturity in *X. multilineatus*, when males stop growing, is influenced by a sex-linked gene (*mc4r*) on the Y chromosome [28]. A tradeoff between survival to and size at sexual maturity helps to maintain the two genetically influenced ARTs via negative frequency dependent selection [29]. In addition, both laboratory [30] and field [25] studies have detected a mortality-growth rate tradeoff for males, where courter males optimize growing faster to increase liklihood of reaching sexual maturity and sneaker males grow slower to optimize living longer as adults. The genetic influences on growth rates have not been identified, however females with courter sires also have faster growth rates compared to females with sneaker sires [31]. In addition, the role of growth rate and ART lineage in relation to female mate preference in *X. multilineatus* is not known.

Building off this body of work, we examined variation in female mate preference for males from the courter ART lineage as compared to males from the sneaker ART lineage, taking into consideration the interactions between female genotype in relation to the male ARTs, growth rates that are influenced by sire's genotype, and social experience. In the first experiment, we manipulated both the composition of the female's social environment and her genotype, where females either experienced 100% sneaker males and had a pure sneaker genotype (sneaker sire, dam with sneaker sire), experienced 100% courter males and had a pure courter genotype (courter sire, dam with courter sire) or experienced a 50/50 split of each tactic and had mixed genotype. In the second experiment, we tested the mate preferences of females from pure sneaker or pure courter genotype that had no sexual experience (virgins), followed by tests of their preferences after we provided them with sexual experience with both types of males. We wanted to examine the influence of sexual experience on strength of preference as we previously demonstrated that virgins are not as choosey, and that their experience with

different frequencies of the male ARTs can influence their strength of preference [27]. In addition, given the mortality-growth rate tradeoff detected previously in this system, we wanted to examine the possibility of disassortative mating in relation to growth rates. While less common than assortative mating, disassortative mating is predicted in cases where traits experience strong balancing selection [32], and could be a mechanism for negative frequency dependent selection [33,34]. This hypothesis predicts that females from the slower growing genotype (sneaker) will have a stronger preference for the faster growing courter males than females from the courter genotype.

## Methods

### Animal care

These experiments were approved by the IACUC of Ohio University (12-L-042). We minimized the welfare impacts on subjects, including not housing adult males together and environmental enrichment of artificial plants and gravel substrate in each individual tank. Every individual tank had an over the back carbon filter to ensure water movement, and remove any water transfer from tank to tank. Bi-weekly water changes occurred in all tanks to additionally support consistent water quality within and across different tanks. We fed all fish Ken's Premium Flake *ad libitum* once daily seven days a week, and brine shrimp *nauplii* (*Artemia* species) *ad libitum* once daily five days a week. Fish were kept on a 12:12-hour light: dark cycle. After the study stimulus males in experiments were returned to their original breeding mesocosms.

### Female mate preference assay

We tested females for their preference for courter males with a standard dichotomous choice test [26] using a 284 L tank dived into three sections partitioned by plexiglass (Fig 1). We marked two additional sections by dotted lines on the outside of the tank within the largest center section. Females were placed into an opaque tube in the center of the tank, and one courter male and once sneaker male were placed on opposite ends of the tank in the outermost

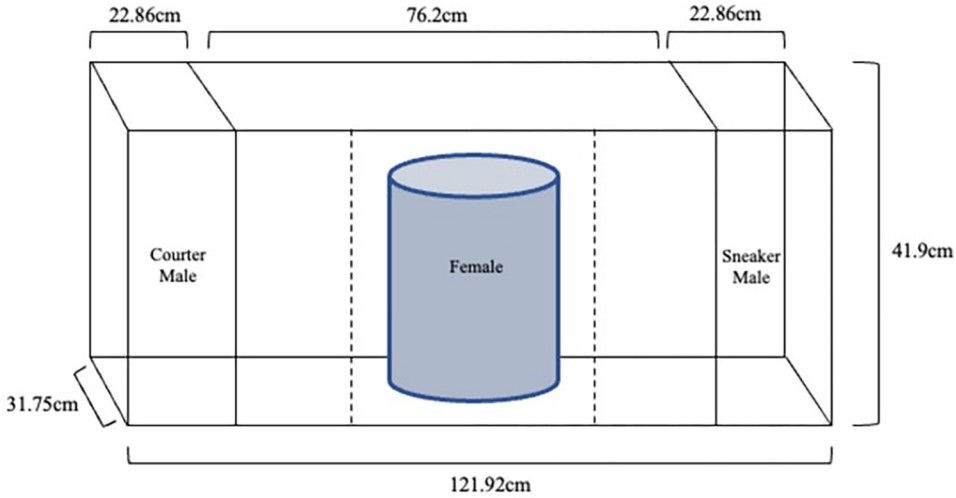

**Fig 1. Schematic of experimental setup for dichotomous choice tank used in preference tests.** Side view with a female located in center compartment and one sneaker and one courter stimulus male located on either the right or the left compartment. All fish were separated by plexiglass indicated by solid lines. Dotted-lines indicate female association zones for each male.

sections. We gave all fish a 10-min acclimation period, after which the female was released and allowed to swim freely. If males/female were inactive during acclimation, the trial was ended and all individuals were returned to their respective "home" tank. Three trials were removed from the experiment and were not included in analyses. While the use of video animations completely controls for variation in male behaviors, there is also a reduction in the strength of female responses [35], which the use of live males allowed us to avoid. The time the female spent in the center zone adjacent to either male (choice zone) was recorded. We made all observations from behind a screen covered with one-way vision film. Each preference test consisted of two 10-minute trials with the side on which the males were placed switched between each trial to control for side bias. After trials were completed, we calculated a female's strength of preference by subtracting the total time spent with the sneaker male from the total time spent with the courter male. Thus, positive preference scores were indicative of a courter male preference and negative values were indicative of a sneaker male preference. We calculated total association time with males as the time spent with the sneaker male plus the time spent with courter male. Higher positive values indicate more time spent overall with males and is an overall indicator of choosiness.

## Experiment one

To examine the effect of a female's genotype and the composition of male alternative reproductive tactics (ARTs) on mate preference we set up six breeding mesocosms as mock communities with an equal ratio of female and males: two mesocosms with equal ratio of both sneaker and courter males (mixed mesocosm), two mesocosms with only sneaker males (sneaker mesocosm), and two mesocosms only with courter males (courter mesocosm). Mesocosm were set up with virgin females and wild caught males and allowed to breed for 4–5 generations prior to testing. Given that the males in this species breed true to ART, and the mesocosms had been breeding for longer than the average female's life span, females from the sneaker mesocosm will therefore be from a pure sneaker genotype (sneaker sire and dam with a sneaker sire), while females from the courter mesocosm will be from a pure courter genotype (courter sire and dam with a courter sire). In addition, females will have had experience with only one of the two types of males based on the community the female came from. We also tested females from mixed mesocosms where females would have more mixed genotypes and experience with both types of males. We tested females from replicate mesocosms for both the pure courter, pure sneaker, and mixed genotypes. Females were photographed and measured in ImageJ [36] 24 hours before mate preference tests to measure standard length.

In total, we tested a total of 32 females from the two courter mesocosms (mean standard length = 33.03 mm; SE ± 0.64, standard length range: 26.58 mm—42.57 mm), 37 from the two sneaker mesocosms (mean standard length = 33.86 mm; SE ± 0.62, standard length range: 25.83 mm– 40.77 mm), and 18 females from the mixed mesocosm (9 early, 9 later, mean standard length = 28.82 mm; SE ± 0.82, standard length range: 24.95 mm– 38.37 mm) for a total of 87 females. Prior to testing, females were isolated in to 22.7-L aquaria for 7–14 days. None of the females dropped fry during this isolation period, suggesting similar stages of gestation [37]. We then tested their mate preferences as described above, using a randomly chosen unique pair of one sneaker stimulus male and one courter stimulus male for each female. We had a total of 20 stimulus males (10 courter, 10 sneaker) and we created 50 unique male pairs for the mate preference assay. Courter males had a mean standard length of 24.83 mm ± SE 0.34 (Range: 32.89 mm– 36.44 mm) and sneaker males had a mean standard length of 27.69 mm ± 0.22 (Range: 26.32 mm– 28.78 mm). The males in a stimulus pair differed in size by 7.09 mm on average (SE = 0.15).

## Experiment two

We tested 20 females from the F2 generation of a paternal half-sibling lab-bred, pedigree population [38]. Each fish had been raised with at least one companion sibling until sexual maturity, at which point the fish were isolated into 18.9-L aquaria with refugia. None of the females in this study were siblings. Each female had a pure genotype (both dam and sire from the same ART lineage) of courter or sneaker ARTs (10 pure courter genotypes, 10 pure sneaker genotypes).

We then preformed a female mate preference assay (as described above) using as stimuli male pairs of one sneaker stimulus male and one courter stimulus male. We had a total of 10 stimulus males (5 courter, 5 sneaker) and created 20 unique male pairs for the mate preference assay. Courter males had a mean standard length of 34.74 mm ± SE 0.61 (Range: 33.63 mm–36.5 mm) and sneaker males had a mean standard length of 28.05 mm ± 0.24 (Range: 27.19 mm– 28.46 mm). The males in a stimulus pair differed in size by 6.84 mm on average (SE = 0.3). Once the females had been tested for their mate preference as virgins, they were given mate experience. The mate experience consisted of placing the female with a unique pair of males that were different that the stimulus males; one sneaker, and one courter male were randomly selected from one of the breeding mesocosm populations within laboratory. One companion female was also randomly selected and added to the tank to limit harassment of the focal female. The companion female's caudal fin was clipped in order to identify her from the focal female. All four fish were placed in to 39-L aquaria with refugia and allowed to socialize freely for four weeks, after which we removed the males and companion female from the tanks. We then repeated the mate preference assay to assess a female's preference for courter males following mate experience. Females were photographed and measured in ImageJ [36] 24 hours before each of the mate preference tests to measure standard length. The two measurements, 28 days apart, were used to calculate growth rates.

## Statistical analysis

For experiment one, we used a non-parametric paired Wilcoxon rank t-test to compare the total time females from each genotype spent associating with the courter versus the sneaker male. We calculated effect size as the absolute of the test statistic divided by the square-root of the sample size. While time spent with courter males (K-S = 0.99 p = 0.77) was normally distributed, time spent with sneaker males (K-S = 0.97 p = 0.024) was not normally distributed.

We examined the factors influencing the strength of female mate preference using a general linear mixed-effect model (GLMM), following a Gaussian distribution in package "nlme" v3.1–137 [39]. The model was checked for multi-collinearity using function vif in the "car" package [40] and normality using a Shapiro-Wilks normality test (p = 0.76). Our GLMM consisted of the following fixed effects that were of primary interest: female standard length (mm), mesocosm (mixed, courter, sneaker) and included replicate mesocosm as a nested factor in our analyses. The specific male pair combination used in the dichotomous choice tank was included as a random effect since the same male pair was repeated in multiple tests. The model was fit using restricted maximum likelihood. We used a Tukey's post hoc test to assess mesocosm treatment relationships when a result was obtained with a p < 0.05. All statistical analysis was done in R software v3.4.1 [41]. We repeated the methods above with a GLMM with the same fixed effects to examine the factors influencing total association time with the males.

For experiment two, we calculated a female's strength of preference for each mate preference trial as the total time spent with the courter male minus the total time spent with the sneaker male. Change in strength of preference was calculated by subtracting a female's virgin strength of preference from their experienced strength of preference. Standard length was

measured in mm twice (measurements 28 days apart) and we calculated growth rate by subtracting the second measurement from the first and dividing by the number of days between each measurement.

We tested for a difference in the strength of preference between inexperienced and experienced females using a Wilcoxon rank t-test. To test for differences in growth rates between females from the two genotypes we used a Welch two sample t-test. We calculated effect size as the absolute of the test statistic divided by the square-root of the sample size. In addition, three scores of female mate preference (strength of preference as virgin, strength of preference as experienced, and change in strength of preference) were examined using three general linear models (GLMs) to evaluate the effect of female genotype (courter vs sneaker), growth rate, and the interaction between female genotype and growth rate. We used the interaction between female genotype and growth rate within the model since we know for this species females with courter sires have a faster growth rate than females with sneaker sires [31]. Graphs presented throughout were created using the "ggplot2" package [42]. Datasets for experiment one (S1) and experiment two (S2) are available in supporting information.

## Results

### Experiment one

Females from the sneaker community type (mesocosm) spent more time with courter males (median: 746s) compared to sneaker males (median: 269s; $p < 0.001$, effect size = 0.95). There was no difference in the time the females from the courter mesocosm spent with courter males (median: 590s) compared to sneaker males (median: 496s; $p = 0.49$, effect size = 0.29). In addition, there was no difference in the time females from the mixed mesocosm spent with courter males (median: 594s) compared to sneaker males (median: 488s; $p = 0.54$, effect size = 0.14).

Strength of mate preference for courter males was explained by mesocosm (GLMM: $\chi 2 = 11.6$ df = 48 p = 0.007), but not female's standard length (GLMM: $\chi 2 = 1.32$ df = 48 p = 0.26). Strength of preference for females from the courter mesocosm was not different compared to strength of preference for the females from the mixed mesocosm (p = 0.83). However, both the courter mesocosm females (p = 0.017) and the mixed mesocosm females (p = 0.025) had a lower strength of preference for courter males compared to the sneaker mesocosm (Fig 2).

Total time spent associating with males was not explained by mesocosm (GLMM: $\chi 2 = 0.64$ df = 48 p = 0.61), and the influence of a female's standard length (GLMM: $\chi 2 = 4.07$ df = 48 p = 0.054).

### Experiment two

We did not detect a change in the strength of mate preference from prior to sexual experience (i.e., virgins) to after sexual experience for females from either genotype (sneaker virgins median = 568.5s, sneaker experienced median = 904.5s, p = 0.19, effect size = 0.41; courter virgins median = 454s; courter experience median = 495s, p = 0.31, effect size = 0.42). The courter females grew faster (0.027 mm/day SE = .0022) than the sneaker females (0.019 mm/day, SE = .002; t = 2.62, p = 0.017), similar to results from previous studies [31].

Strength of preference in virgin females was unrelated to female genotype (GLM: t = 1.22 df = 16 p = 0.24), growth rate (GLM: t = 0.85 df = 16 p = 0.41), or the interaction between female genotype and growth rate (GLM: t = -1.3 df = 16 p = 0.22). Strength of preference for experienced females was influenced by a female's genotype (GLM: t = 2.53 df = 16 p = 0.02), an interaction between growth rate and genotype (GLM: t = -2.34 df = 16 p = 0.033) but unrelated to growth rate overall (GLM: t = 1.24 p = 0.23). Similar to the results for the females from the mesocosms, sneaker females had a stronger preference for courter males as compared to

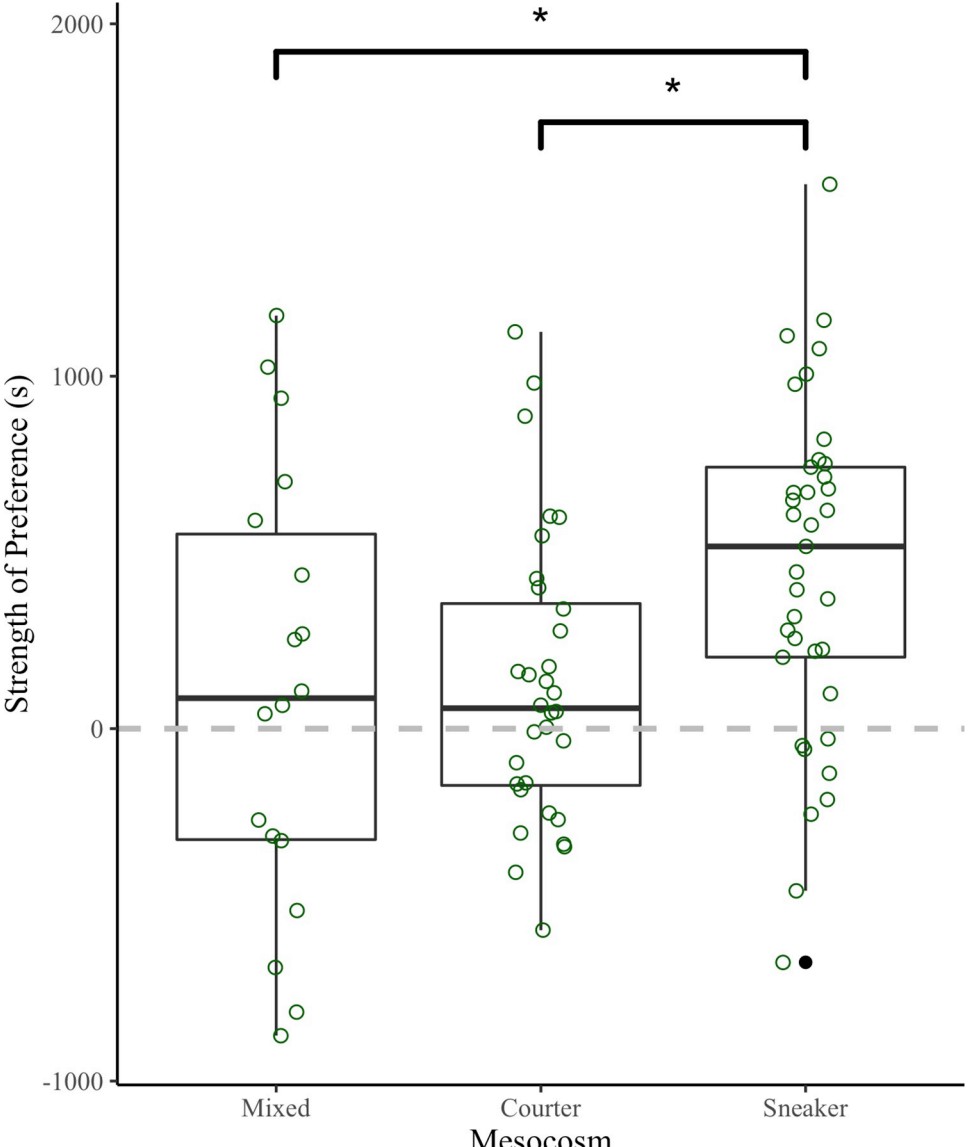

**Fig 2. The strength of preference (s) for courter males from the different mesocosm treatments: Mixed (n = 18), courter (n = 32), and sneaker (n = 37).** * = significance difference where p < 0.05.

courter females after mating experience (Fig 3). There was an interaction between growth rate and genotype when explaining strength of preference (GLM: t = -2.34 p = 0.033, Fig 4), with faster growth rates correlated with a decrease in strength of preference in sneaker females and an increase in strength of preference in courter females.

Change in preference when females were inexperienced as compared to experienced was unrelated to female genotype (GLM: t = -0.42 p = 0.68), growth rate (GLM: t = -0.02 p = 0.98), or the interaction between genotype and growth rate (GLM: t = 0.27 p = 0.79).

## Discussion

A female's genotype in relation to the male alternative reproductive tactic (ART) (courter lineage or sneaker lineage) affected the strength of her mate preference for the ARTs in

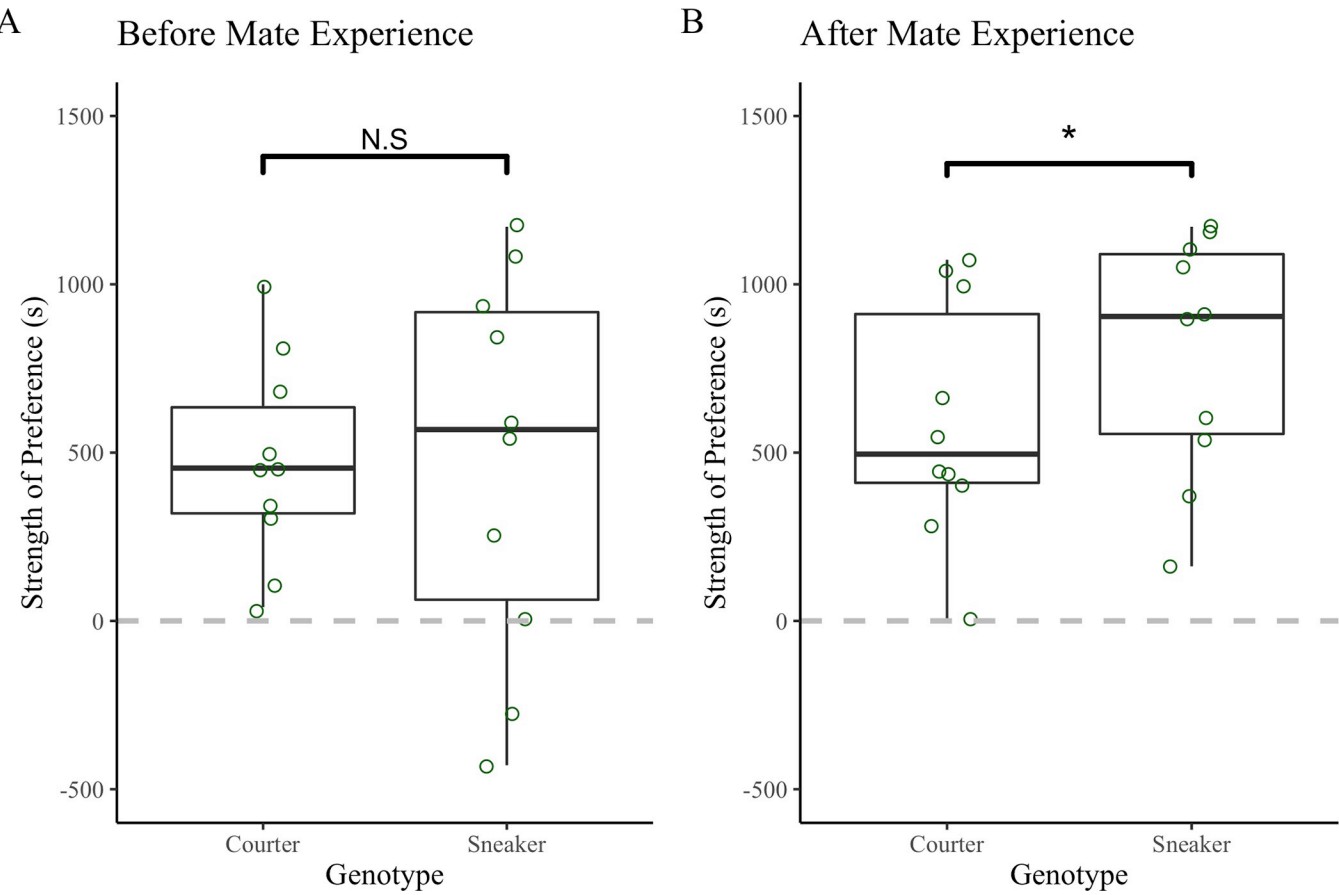

**Fig 3. Strength of preference (s) of females from two different genotypes (courter vs. sneaker).** A.) Female's strength of preference before mate experience B.) Female's strength of preference after mate experience with each male tactic. * = significance difference where p < 0.05.

*Xiphophorus multilineatus*. This pattern was detected in females from breeding mesocosms (experiment one) and females with known pedigrees (experiment two) after mating experience, regardless of whether experience was with only one tactic or both. Sneaker females that have slower growth rates had a stronger preference for the faster growing courter males than courter females. In addition, variation in strength of preference was influenced by a female's growth rate, but not in the same direction for the two female genotypes. Within sneaker females, strength of preference for the larger courter males decreased with increased female growth rates, while within courter females strength of preference trended towards increase with increased female growth rates. Below we consider why mating experience may be necessary for detecting the differences between the genotypes in strength of preference, as well as potential mechanisms and hypotheses that could explain the variation in strength of preference based on genotype and growth rates. Given the known growth rate differences between the ARTs that are associated with a mortality-survival tradeoff in this system, we present a hypothesis suggesting that some of the variation in mate preferences in this system could be due to disassortative mate preferences in relation to growth rates.

The differences in strength of preferences between the two genotypes of females was detected in females from the mesocosms that had mating experience, and in the pedigree females but only once they were given mating experience. This result concurs with previous findings in this species, that virgin females with no mating experience have weaker preferences

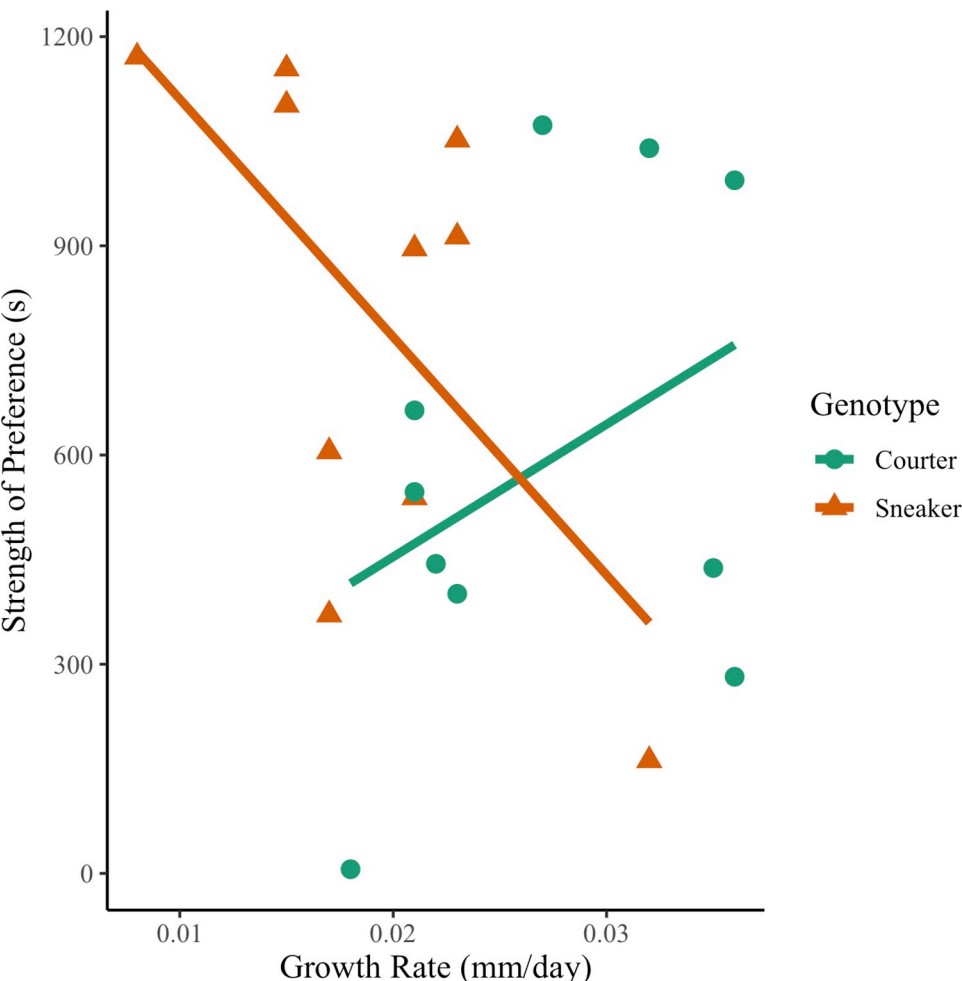

**Fig 4. Interaction between growth rate (mm/day) and females genotype (courter vs. sneaker) on strength of preference (s).** Green line and circles indicate courter genotype and red line and triangles indicate sneaker genotype.

for courter males than females with mating experience [27]. As experience interacting with males influenced mate preferences, one possible explanation for the differences found between the experienced courter and sneaker females could be differences in costs or benefits of socially learning a mate preference. Learning has been examined in this species, and females with bigger brains learned faster, as well as courter females learned faster than sneaker females [43]. However, we did not find that the change in preference after gaining mating experience was explained by genotype. Given that brains are energetically expensive, we expect that further study of the tradeoffs between brain size, growth rates, and fecundity may provide increased insights into the variation across the genotypes in response to mating experience that we detected. Finally, swordtail females store sperm [44] and aspects of sperm competition and longevity can differ among the ARTs [45]. Both are additional factors that could be examined in relation to the influence of sexual experience on mating preferences.

The relationship we found between strength of preference and growth rate, which was different depending on a female's genotype, has an interesting connection to previous work on female mate preferences and cortisol levels in this species. Courter females were found to have higher cortisol levels, and the strength of female mate preferences were associated with higher

cortisol levels [46]. Cortisol is known to be associated with reduced growth, and stress responses in fish [47,48], and therefore cannot explain the faster growth rates we detected in the courter females. However, cortisol differences could explain the differences in strength of preference between the genotypes, as well as the reduction in mate preference in the sneaker females as their growth rates increased. Examining this possibility will require further study, but will be particularly important to consider for hypotheses that attempt to explain the variation in mate preferences in relation to growth rates in this species.

Given that costs to mate preferences have been well documented [6,20] it is important to consider their potential impact on the relationships we detected between a female's genotype, growth rate and strength of mate preference. Costs can range from energy invested while searching for mates, to tradeoffs with other life history traits like growth, to the sensory systems necessary to detect differences between mates. We assessed one of costs of being choosey by measuring the total time females associated with the males during testing (not the relative time spent with the two stimulus males), but detected no influence of a female's genotype. However, if being choosey requires increased investment in sensory processing, such as larger brains, this could result in less energy available for growth and could explain why the genotype with slower growth (sneaker) had stronger preferences. Results from previous work in this system found the opposite relationship between brain size and growth, as the faster growing courter females were shown to learn faster and have larger brains than sneaker females [49]. Costs are likely to play an important role in the variation in mate preferences in most systems and further assessment of the costs in this system will be important in relation to testing hypotheses for evolution of these mate preferences [50]. Females can store sperm in these fishes [44], leading to additional arguments for examining both increased and decreased costs of mate preferences. Males provide no resources to females or offspring in swordtails, suggesting females may prefer larger males because male size is a proxy for some aspect of mate quality that provides indirect benefits—offspring inheritance of adaptive traits. Given that females showed variation in their preferences for the male ARTs and differences between the male ARTs in *X. multilineatus* have been studied extensively, this is an excellent system in which to further examine the adaptive benefits of female mate preference.

Evolutionary hypotheses for the variation in mate preferences we detected will want to consider known differences between the male ARTs in this system. ARTs in *X. multilineatus* are dimorphic not only for body size and shape [22,51], but also plasticity in mating behaviors [22], age at sexual maturity when males stop growing [23], and sperm longevity [46]. Variation in the *Mc4r* gene on the Y-chromosome has been found to be associated with some these traits [24,28], but is unlikely to explain all the genetic variation and tactical dimorphism between the two ARTs. The two ARTs are maintained by frequency dependent selection and a tradeoff between the benefits of being larger and gaining more mating success (optimized by the courter males), and the benefits of a reaching sexual maturity sooner at a smaller size, which increases the probability of reaching sexual maturity (optimized by the sneaker males) [29]. Interestingly, the ARTs are also dimorphic for growth rate. Measured in the laboratory, the larger courter males grow faster than the smaller sneaker males [24]. This difference in growth rates leads to an additional tradeoff: juvenile growth rates and adult mortality (i.e. mortality-growth rate tradeoff, [52], which has been documented for *X. multilineatus* males in both the laboratory [30] and the field [25]. Assuming genetic variation is at least in part responsible for the differences in growth rates between female's based on genotype (courter versus sneaker lineage), disassortative mating based on growth rates would provide offspring with benefits by optimizing the mortality/growth rate tradeoff. To further develop this hypothesis, it will be important to examine the heritability of growth rates, teasing apart the contribution of dam versus sire genotype to offspring growth rate. In addition, it would be interesting to examining

the variation in female mate preference for male size within courter males, as there is still a relationship between male size and growth rates, but not the additional differences in other traits (e.g. behavior and mating success) found between the alternative reproductive tactics.

Finally, two aspects of our results do not match with previous studies of mate preference for the courter male ART by wild-caught females from this species: we did not detect a significant preference in the females from the mixed or courter only mesocosms, and we did not detect an influence of female size on this preference. One possibility is that we need a better understanding the factors that influence variation in female size, in addition to the relationship between growth rates and female size in the laboratory as compared to wild-caught females which are not likely to be the same. We have recently increased our ability to assess the growth rates of wild-caught females using otoliths [25], and therefore future studies can examine the relationship between size, growth rates and preferences in wild-caught females to determine if the pattern we detected in lab reared females is the same for wild-caught females.

## Conclusion

Variation in female mate preferences in *Xiphophorus multilineatus* have been implicated in the negative frequency dependent selection that maintains the male ARTs in this species [53], and found to correlate with female size [27] as well as levels of cortisol [46]. Here we document variation in relation to a female's genotype and growth rate, suggesting that females with faster growth rates (courter lineage compared to sneaker lineage, and variation within sneaker genotype) have weaker preferences for the faster growing courter males. Neither the sensory bias hypothesis for larger males nor a hypothesis based on the costs to being choosey can adequately explain our results. Given what is known about the ARTs in this system, we suggest an additional hypothesis warrants further investigation. The growth rate optimization hypothesis suggests that when choosing between potential mates, females are selected to balance the mortality/growth rates tradeoff by providing their offspring with genotypes from their sires genotypes will optimally balance growing fast prior to sexual maturity (increased probability of reaching sexual maturity, larger adult size) with the costs of growing faster due to a shorter adult lifespan. While not the first hypothesis to suggest a relationship between behaviors and mortality-growth rate tradeoffs [54], our hypothesis is the first to suggest variation in female mate preferences have evolved in response to producing offspring that optimize these tradeoffs. In addition, recent work on the selection regimes and the genetic architecture favoring the emergence of disassortative mating [32] provides a theoretical framework in which to test this hypothesis. Studies of variation in female mate preferences, especially in systems with alternative reproductive tactics, can help disentangle the multiple influences on mate preferences and lead to a better understanding of their evolution. In this system, further investigation of the genetic influences on growth rates, selection on growth rates in females, and differences in the optimal growth rates between males and females are needed. Further study of the complex relationship between male size and growth rates [55] has the potential to provide additional support across taxa for the hypothesis that mate preference for male size may be under selection due to size being indicative of growth rate.

## Supporting information

**S1 Dataset. Dataset for experiment one.**
(PDF)

**S2 Dataset. Dataset for experiment two.**
(PDF)

## Acknowledgments

We would like to thank John Bobo for support in conducting behavioral trials in experiment two in this study, Hannah Griebling for providing us with the females to test both before and after mating experience, and for the students that helped maintain the fish in the Morris laboratory with safe animal care practices.

## Author Contributions

**Conceptualization:** Meredith Fitschen-Brown, Molly Morris.

**Data curation:** Meredith Fitschen-Brown, Molly Morris.

**Formal analysis:** Meredith Fitschen-Brown.

**Funding acquisition:** Molly Morris.

**Investigation:** Meredith Fitschen-Brown, Molly Morris.

**Methodology:** Meredith Fitschen-Brown, Molly Morris.

**Project administration:** Meredith Fitschen-Brown, Molly Morris.

**Resources:** Meredith Fitschen-Brown, Molly Morris.

**Software:** Meredith Fitschen-Brown.

**Supervision:** Meredith Fitschen-Brown, Molly Morris.

**Validation:** Meredith Fitschen-Brown, Molly Morris.

**Visualization:** Meredith Fitschen-Brown.

**Writing – original draft:** Meredith Fitschen-Brown, Molly Morris.

**Writing – review & editing:** Meredith Fitschen-Brown, Molly Morris.

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
