## [Decision Letter · Decision Letter 0]

11 Apr 2023

PONE-D-23-01131Genotype and growth rate influence female mate preference in Xiphophorus multilineatus: potential selection to optimize mortality-growth rate tradeoffPLOS ONE

Dear Dr. Fitschen-Brown,

Thank you for submitting your manuscript to PLOS ONE. After careful consideration, we feel that it has merit but does not fully meet PLOS ONE’s publication criteria as it currently stands. Therefore, we invite you to submit a revised version of the manuscript that addresses the points raised during the review process. Please submit your revised manuscript by May 26 2023 11:59PM. If you will need more time than this to complete your revisions, please reply to this message or contact the journal office at plosone@plos.org. Please include the following items when submitting your revised manuscript:A rebuttal letter that responds to each point raised by the academic editor and reviewer(s). You should upload this letter as a separate file labeled 'Response to Reviewers'.A marked-up copy of your manuscript that highlights changes made to the original version. You should upload this as a separate file labeled 'Revised Manuscript with Track Changes'.An unmarked version of your revised paper without tracked changes. You should upload this as a separate file labeled 'Manuscript'.

We look forward to receiving your revised manuscript.

Kind regards,

Quenton M. Tuckett

Academic Editor

PLOS ONE

Journal Requirements:

https://journals.plos.org/plosone/s/file?id=ba62/PLOSOne_formatting_sample_title_authors_affiliations.pdf"

“This project was supported in part by the Ohio Center of Ecological and Evolutionary Studies graduate research fellowship program and a Baker Grant from Ohio University.”

“This project was supported in part by the Ohio Center of Ecological and Evolutionary Studies graduate research fellowship program and a Baker Grant from Ohio University.

“We note that you have provided additional information within the Acknowledgements Section that is not currently declared in your Funding Statement. Please note that funding information should not appear in the Acknowledgments section or other areas of your manuscript. We will only publish funding information present in the Funding Statement section of the online submission form.

“This project was supported in part by the Ohio Center of Ecological and Evolutionary Studies graduate research fellowship program and a Baker Grant from Ohio University.”

Additional Editor Comments:

Dear authors,

First, let me apologize for the delay in getting this decision finalized; it was partly my doing and also an inability to located suitable reviewers. That said, we now have a decision based on two reviews of your manuscript "Genotype and growth rate influence female mate preference in Xiphophorus

multilineatus: potential selection to optimize mortality-growth rate tradeoff". Both reviewers indicated the manuscript could be suitable for publication, but it will require major revisions. Please find the detailed comments from both reviewers below.

In particular, consider how the results are explained, including consideration of alternate hypotheses; this was identified by both reviewers.

Sincerely, Quenton M. Tuckett

Reviewers' comments:

Reviewer's Responses to Questions

**Comments to the Author**

1. Is the manuscript technically sound, and do the data support the conclusions?

Reviewer #1: Partly

Reviewer #2: Yes

2. Has the statistical analysis been performed appropriately and rigorously? 

Reviewer #1: Yes

Reviewer #2: Yes

3. Have the authors made all data underlying the findings in their manuscript fully available?

Reviewer #1: Yes

Reviewer #2: Yes

4. Is the manuscript presented in an intelligible fashion and written in standard English?

Reviewer #1: Yes

Reviewer #2: Yes

5. Review Comments to the Author

Reviewer #1: This MS deals with a study of a model system that has been detailed studied in the past for mate choice, the live-bearing fish Xiphophorus multilineatus. Previous studies have shown that males have genetically-based alternative reproductive tactics (ARTs): one male type is called COURTER (C type; having higher size and growth rate, later sexual maturity and displaying extreme courtship behaviour) while the other is called SNEAKER (S type; having smaller size, lower growth rate and earlier sexual maturity, with earlier courtship and coercive force copulation). Natural populations show a possible protected polymorphism in ARTs via presumably frequency dependent selection. Although much is known, the details of the mechanism acting as well as the different interactions between traits (size, growth rate, mate choice tactic) are basically unknown. Therefore, the study provides two new experiments trying to improve the knowledge on the system. The first experiment, a female choice design, try to test if the alternative reproductive genotypes (i.e. genotypes) show a pattern of assortative mating or not. The dependent variables were preference (female time expend with C – female time expend with S) and choosiness (female time expend with C + female time expend with S). The studied treatment were three genotypes (C; S and both included) in (non-virgin) females and two (C and S) in males. As the genotypes cannot be morphologically identified, they produced them by rearing specimens of each phenotype for 4-5 generations in mesocosms. The results show a trend consistent with negative assortative mating (a mechanism able to cause negative frequency-dependent sexual selection). The second experiment tested the mating preferences of 10 pure (F2) female genotypes with males of both genotypes as virgin females, then they got some mate experience (maintained 4 weeks in presence of another female plus two males) and them tested again for mating preference as before. As females were measured at both dates they could calculate female growth rate as well. This second experiment is important for understanding the results of the first experiment as well as identifying trait interaction contributing to mate choice. The preference showed a significant interaction between genotypes and growth rate (positive relationship in C and negative in S) a trend which do not support any of the former hypothesis used in this context. The authors discuss and speculate why they got these results as well as their possible interpretation. This study adds a useful piece of information to the study of mate choice in this system, it is reasonable well designed and analysed and so its merits publication. However, I found several potential drawbacks that should be discussed or even mentioned/discussed in the text as potential limitations of the study. I list them below

1. The analyses were done using several fishes repeated in different trials. For example, in experiment 1 they used 10 C and 10 S males, for an overall 87 trials. I understand that they tried to use different combinations of males in different trials (50 new, and so at least 40% of data was not independent). Actually, perhaps it is not bad to use the same males for different female genotypes, providing that they behaviour (of the males) do not change with experience (but see second experiment). I do not think that this invalidates the whole experiment but the potential consequences and limitations of this should be discussed. The same happened in experiment 2.

2. The authors claim that they got a results that supports negative assortative mating (lines 266-269). Actually I think that they got complex results. I will discuss results for both experiments.

2.1. Although in experiment 1 S female prefers to mate the other type, it did not happen with the C females. Notice that a pure negative assortative mating trend will require of both genotypes favouring the other or alternative that the four combinations of possible mates show a sexual isolation index that was negative, or similar. So the authors should be more caution with this result, I think.

2.2. Similarly, in the second experiment, S type female showed a trend consistent with growth rate favouring disassortative mating, as claimed in the text (lines 369-372). Again this was far from being so clear, I think, as the relationship is so in the S genotype but it is the opposite in the C one. Again the authors should be caution with the interpretation regarding assortative mating as they got complex results.

3. One interesting result has not achieved enough author attention, I think. They got that ART trait (their genotypes) show actually a plastic characteristic, as the trait needs of certain previous experience in order to produce certain aspects of the mate choice, as the preference for particular genotype needs of previous experience. I do not know if this has been observed before in other organism, and perhaps merits a particular discussion about the interaction between genetic determination and phenotypic plasticity, aspects considered two alternatives but which could also be two sides of the same coin.

4. The authors and too worried about to produce an evolutionary hypothesis/mechanism that explain their results, which can be understandable, but their new hypothesis seems to me too speculative, perhaps is too soon to advance new hypothesis and just more clear and compressive data is just needed. In my opinion, they should try to reduce the importance of the speculative ideas in discussion, and emphasize the new findings in relation to the previous ones, without any need to present a closed interpretation.

5. In addition, I have a few comments on the text

. Lines 17-19. They only describe the results of one genotype, say that the other showed the opposite at least.

. Lines 63-64. Too frequent repeat “females with courter sires” Once you explained how you got these females they are “courter females”. I think that some sentences could be simplified in the text.

. Line 64-65. It should be explained the exact characteristic of the mate choice relationship in guppy

. Line 81-82. Perhaps this explanation is obvious for the authors or specialized readers, but I think that general reader needs a better explanation about how such hypothesis works and produces its predictions.

. Lines 85-87. Actually several theoreticians claim that disassortative mating may directly cause negative frequency-dependent sexual selection, and so disassortative mating would be the cause of the selection not the consequence (Pusey and wolf 1996. TREE, 11: 201-206; Hedrick several papers; for example 2016; Evolution 70: 757-766). Any fast search on the WEB should give a few more useful references to discuss this if interested.

. Lines 132-134. I personally do not see clear the test group factor used, for me repeated measurements are typically analysed as a nested factor instead of a new factor, as you are not interested per se in such factor just to know if the main effects are true irrespective of such variation.

. Legend of Figure 2. The P > 0.05, I assume that actually means <

I enjoined the MS and so hope that authors may be able to produce an improved version.

Reviewer #2: The authors of “Genotype and growth rate influence female mate preference in Xiphophorus

multilineatus: potential selection to optimize mortality-growth rate tradeoff” report on two experiments used to identify the factors affecting female choice for males exhibiting alternative reproductive tactics. Overall, I enjoyed the manuscript; it wasn’t overly long and rarely strayed from the main objectives. There are some writing issues throughout, but always minor. I do note one issue that should be mentioned alongside their chosen hypotheses: the potential for sperm storage and quality to affect the results. See below for one specific issue (sperm storage) and some specific comments listed line by line.

Specific Issues

1. Somewhere the authors should discuss sperm storage and competition might affect the results of this experiment and also how it might be his might be related to their hypothesis on 272-275. For example, there has been studies on X. nigrensis showing that male tactic is related to traits related to sperm competition (e.g., sneakers might have sperm with greater viability and longevity compared to courters; doi: 10.1098/rsbl.2011.0286). I also think a discussion of sperm storage (in general) would be warranted early on.

Specific Comments

L36: comma after physiology

L93: “not hosing”

L93-94: I am curious about how these sterile environments might affect mate choice experiments

L95: any water quality testing?

L104: remove “a”

L102-: regarding the dichotomous choice tank; this is a bit unclear without a diagram. In doing an image search, these tanks will seemingly vary

L124: “mesocosms”

L141: lowercase females

L141: 22.7-L aquaria

L144: delete “number of”

L146: more important than reporting the range for both males and females would be to examine if there are differences

L156: delete “number of”

L157: once again; differences between tested populations would be more informative, even if expected (e.g., sneaker and courter)

L161: “experience males”?

L208: how was effect size calculated? Was this presented in the methods?

L218: sometimes test statistics and df are reported and at other times not

L228: delete “from”

L239: some variation in significant figures

L257-260: that’s a mouthful

L297: comma after “growth rates”

L299-312: not quite convinced by this explanation and not quite sure why it was included. Some other explanations might be more fruitful. For example, why not mention sperm traits and alternative reproductive tactics? This seems like a missed opportunity, perhaps even something that should be discussed.

L315-316: is this reference comparable to a swordtail that exhibits sperm storage? I’m not sure; I am also not sure how much or how little sperm storage affects the costs of assessing and choosing mates. Presumably, if females can store the sperm from multiple males, there could be arguments for both an increase and decrease in costs.

L367: a possible place for sperm quality as well?

6. PLOS authors have the option to publish the peer review history of their article (what does this mean?). If published, this will include your full peer review and any attached files.

Reviewer #1: **Yes: **Emilio Rolán-Alvarez (Universidade de Vigo, Spain)

Reviewer #2: No

---

## [Author Response · Author response to Decision Letter 0]

24 May 2023

Additional Editor Comments:

In particular, consider how the results are explained, including consideration of alternate hypotheses; this was identified by both reviewers.

Alternative hypotheses and additional explanation of results were addressed in the discussion section. Additional information can be found below in the responses to the reviewer’s comments. 

Reviewers' comments:

Reviewer #1: 

1. The analyses were done using several fishes repeated in different trials. For example, in experiment 1 they used 10 C and 10 S males, for an overall 87 trials. I understand that they tried to use different combinations of males in different trials (50 new, and so at least 40% of data was not independent). Actually, perhaps it is not bad to use the same males for different female genotypes, providing that they behaviour (of the males) do not change with experience (but see second experiment). I do not think that this invalidates the whole experiment but the potential consequences and limitations of this should be discussed. The same happened in experiment 2.

For experiment 1, we attempted to control for the re-use of male pairs by including it as a random repeated effect in the model. Especially since male pairs in experiment 1 were repeated with different females. Within, the second experiment we only had unique pairs of males presented to the females, therefore we did not include it as a random effect in the model. Additionally, males that were inactive (sat on the bottom of the tank) or did not present courtship behavior (sigmodal displays) were not used for a trial and this was determined during the first acclimation period. This information has now been included in methods section (line 109). However, there is no way to insure exact behavioral consistency across trials without using video animations or mechanical models. While this is one of the drawbacks of using live males, previous work has found that using live males and video animations gave similar overall results, however there was a reduction in the strength of female responses using videos (which in the current study we chose to avoid). This point was included in the methodology section (line 115-117). 

2. The authors claim that they got a results that supports negative assortative mating (lines 266-269). Actually I think that they got complex results. I will discuss results for both experiments.

2.1. Although in experiment 1 S female prefers to mate the other type, it did not happen with the C females. Notice that a pure negative assortative mating trend will require of both genotypes favouring the other or alternative that the four combinations of possible mates show a sexual isolation index that was negative, or similar. So the authors should be more caution with this result, I think.

Yes, we agree that we did not detect what the reviewer is defining as “pure” negative assortative mating. While we detected significant differences in the strength of preference between the female genotypes, with a stronger preference for the faster growing males in the sneaker females that had slower growth, no preference was detected in the faster growing courter females. In addition, for experiment 2 where we had individual female growth rates and could examine the variation within the females of a genotype, the relationship was only found in one of the two genotypes (the females with the slower growth rate), but was different from the relationship for the courter genotype females. Therefore, we agree that the pattern is more complex than pure negative assortative mating, and have changed our language throughout the discussion to reflect this. 

2.2. Similarly, in the second experiment, S type female showed a trend consistent with growth rate favouring disassortative mating, as claimed in the text (lines 369-372). Again this was far from being so clear, I think, as the relationship is so in the S genotype but it is the opposite in the C one. Again the authors should be caution with the interpretation regarding assortative mating as they got complex results.

Discussion was redesigned to more accurately represent the complexity of the results, and not just focus on the disassortative mating hypotheses. 

3. One interesting result has not achieved enough author attention, I think. They got that ART trait (their genotypes) show actually a plastic characteristic, as the trait needs of certain previous experience in order to produce certain aspects of the mate choice, as the preference for particular genotype needs of previous experience. I do not know if this has been observed before in other organism, and perhaps merits a particular discussion about the interaction between genetic determination and phenotypic plasticity, aspects considered two alternatives but which could also be two sides of the same coin.

Yes, this is an interesting result, and one that is the focus of a recent meta-analysis (Richardson and Zuk 2023). While this review did not find that across studies virgin females were less choosey, most studies do not examine or at least report both as we do here. We included a paragraph specifically on the role of experience acting on genotypes and the role within mate preference and mate choice (see paragraph starting on line 340). 

Richardson, J., & Zuk, M. (2023). Unlike a virgin: a meta-analytical review of female mating status in studies of female mate choice. Behavioral Ecology, 34(2), 165-182.

4. The authors and too worried about to produce an evolutionary hypothesis/mechanism that explain their results, which can be understandable, but their new hypothesis seems to me too speculative, perhaps is too soon to advance new hypothesis and just more clear and compressive data is just needed. In my opinion, they should try to reduce the importance of the speculative ideas in discussion, and emphasize the new findings in relation to the previous ones, without any need to present a closed interpretation.

Discussion was redesigned to focus more heavily on findings surrounding social experience and female’s genotype on strength of preference for courter males. We do still present discussion of hypotheses to potentially explain the influence of growth rates on preferences we detected. Judy Stamps (2007) presents clear reasoning/arguments for growth-mortality tradeoffs, and we have found support for these tradeoffs in our fish in both the laboratory and the field (line 602-603). We think that the influence this tradeoff may have on mate preferences is something that future scientists should investigate not only in our system, but in general. However, we have paired the discussion of the potential influence of this tradeoff on mate preferences to a paragraph within clear statements of need for future research. 

5. In addition, I have a few comments on the text

Lines 17-19. They only describe the results of one genotype, say that the other showed the opposite at least.

Line 17-19 edited sentences to state:

In addition, the relationship between strength of preference and growth rate depended on a females’ genotype; females with sneaker genotypes decreased their preference as their growth rates increased, a pattern that trended in the opposite direction for the females from the courter genotypes.

Lines 63-64. Too frequent repeat “females with courter sires” Once you explained how you got these females they are “courter females”. I think that some sentences could be simplified in the text.

After methods section changed females with courter sires to just courter females/sneaker females for the rest of the manuscript. 

Line 64-65. It should be explained the exact characteristic of the mate choice relationship in guppy

The preference for male body size, as the study we cited specifically found preference for body size in that population of guppies (line 68-69). However, we ended up taking the citation out as we restructured the paragraphs to address other points. 

Line 81-82. Perhaps this explanation is obvious for the authors or specialized readers, but I think that general reader needs a better explanation about how such hypothesis works and produces its predictions.

Thank you for noting that this prediction was not clear. Even preferences for genetic benefits can be plastic in relation to experience (reduced for example if costs are high on being choosey). We have restated our interest in examining virgin and experienced females (line 88).

Lines 85-87. Actually several theoreticians claim that disassortative mating may directly cause negative frequency-dependent sexual selection, and so disassortative mating would be the cause of the selection not the consequence (Pusey and wolf 1996. TREE, 11: 201-206; Hedrick several papers; for example 2016; Evolution 70: 757-766). Any fast search on the WEB should give a few more useful references to discuss this if interested.

Thank you for these references. We added some additional reverences here (see lines 94-96) and throughout the manuscript in relation to this hypothesis. 

Lines 132-134. I personally do not see clear the test group factor used, for me repeated measurements are typically analysed as a nested factor instead of a new factor, as you are not interested per se in such factor just to know if the main effects are true irrespective of such variation.

We changed the model to using test group as a nested factor instead of a covariate/new factor. We agree with this interpretation of the test group factor since our focus was to understand if mesocosm or standard length influenced strength of female mate preference. We included this in both the methods and changed the results. However, the outcome did not change when using test group as a nested factor. We also changed the test group terminology to replicate to be more clear. 

Legend of Figure 2. The P > 0.05, I assume that actually means <

Changed Figure 1 & 2 legend mistake. 

I enjoined the MS and so hope that authors may be able to produce an improved version.

Thank you, and we really appreciate the comments and feedback provided as well as the time spent on improving our work. 

Reviewer #2: 

1. Somewhere the authors should discuss sperm storage and competition might affect the results of this experiment and also how it might be his might be related to their hypothesis on 272-275. For example, there has been studies on X. nigrensis showing that male tactic is related to traits related to sperm competition (e.g., sneakers might have sperm with greater viability and longevity compared to courters; doi: 10.1098/rsbl.2011.0286). I also think a discussion of sperm storage (in general) would be warranted early on.

We are aware of evidence for influence of sperm competition on male mate preferences in guppies (Dosen and Montgomerie 2003) but are unclear on how to evaluate the influence of sperm storage and/or competition in our case (female mate preferences). We did not quantify if females had actually mated with males. While they were in a social environment and exposed to males for a long period of time, we assume that courtship and mating (or at least attempted mating) occurred at some point. For this reason, we cannot disentangle in this study social dynamics from copulation. However, the differences in longevity of the sperm and sperm competition in relation to the ARTs are interesting aspects of this system that we have now included in the discussion of the influence of experience on mate preferences. 

Also, Line 166 we added a statement that we did not have any individuals drop fry during or after testing. Therefore, we assumed that females were more or less at the same gestational stage during experimentation, as this could potentially impact the results of the mate preference trials as well (Ramsey et al. 2011). 

Dosen, L. D., & Montgomerie, R. (2004). Female size influences mate preferences of male guppies. Ethology, 110(3), 245-255.

Ramsey, M. E., Wong, R. Y., & Cummings, M. E. (2011). Estradiol, reproductive cycle and preference behavior in a northern swordtail. General and comparative endocrinology, 170(2), 381-390.)

Specific Comments

L36: comma after physiology

Added comma

L93: “not hosing”

Fixed grammar error to not housing 

L93-94: I am curious about how these sterile environments might affect mate choice experiments

Absolutely, we are too. While we have not identified the factors influencing the differences, we discuss how the environment of the lab as compared to the wild may influence mate preferences in the paragraph starting on line 421

L95: any water quality testing?

Larger Tanks (50-100 gallons): Continuous monitoring of ammonia with Seachem patch, 

Weekly of nitrates and pH levels. Smaller tanks (5-10 gallons): ammonia, nitrates and pH levels tested once every 3 months. This information was added to the animal care section.

L104: remove “a”

Removed “a”

L102-: regarding the dichotomous choice tank; this is a bit unclear without a diagram. In doing an image search, these tanks will seemingly vary

Agreed, a schematic was included as a new figure to show both the set up and dimensions of the dichotomous choice thank that we use in the lab. 

L124: “mesocosms”

Switched aquarium experiments to breeding mesocosm experiments 

L141: lowercase females

Changed females to lowercase

L141: 22.7-L aquaria

Changed

L144: delete “number of”

Deleted

L146: more important than reporting the range for both males and females would be to examine if there are differences

In this line we were reporting the range in the sizes between courter and sneaker males that were used as stimuli in the mate preference trials. However, we agree that it would be useful to present the size differences of the pairs that were presented to the female. This information is now included.

L156: delete “number of”

Deleted

L157: once again; differences between tested populations would be more informative, even if expected (e.g., sneaker and courter)

Differences between the sizes of the males in the stimulus pairs is now presented (lines 172-173).

We apologize that this was not clear. This range was referring to the courter stimulus males and sneaker stimulus males and was not actually describing any of the populations. 

L161: “experience males”?

We want the readers to understand that these males were different males from the stimulus males used during the mate preference trails. Therefore, the females had never seen the stimulus males before being tested. We have clarified this point in the text.

L208: how was effect size calculated? Was this presented in the methods?

Effect size calculation was added to the statistical methods section.

L218: sometimes test statistics and df are reported and at other times not

We have standardize based on the statistical analysis we did both the test statistics and df being reported. 

L228: delete “from”

Deleted

L239: some variation in significant figures

Added variation in individuals to boxplots via the jitter function in ggplot2. 

L257-260: that’s a mouthful

Paired sentence down to make more straightforward. Got rid of preference for courter males because that was the same throughout the wholes study. 

L297: comma after “growth rates”

Added comma

L299-312: not quite convinced by this explanation and not quite sure why it was included. Some other explanations might be more fruitful. For example, why not mention sperm traits and alternative reproductive tactics? This seems like a missed opportunity, perhaps even something that should be discussed.

We included the cortisol as a hypothesis for the relationship between growth rate differences and mate preferences, as we previously detected an association between cortisol and strength of mate preferences between the genotypes. In addition, as cortisol has been demonstrated to be associated with growth rates in addition to stress, it has the potential to be a mechanism influencing the variation we detected in both growth rates and preferences between the genotypes. 

We are still unclear how sperm traits in this system might play a role in female mate preference and genotype at this time. However, the differences in the ARTs in growth rates is one of the factors that drives the hypothesis for disassortative mating (females from sneaker lineage with slower growth rates having a stronger preference for courter males with faster growth rates). 

We reduced focus on the physiological and evolutionary future hypotheses in the discussion, and includ sperm trait differences in the ARTs in swordtails as an avenue of future research in relation to female mate preference. 

L315-316: is this reference comparable to a swordtail that exhibits sperm storage? I’m not sure; I am also not sure how much or how little sperm storage affects the costs of assessing and choosing mates. Presumably, if females can store the sperm from multiple males, there could be arguments for both an increase and decrease in costs.

We have modified this paragraph on costs. Additional references have been included. Sperm storage is an interesting aspect of the system in relation to costs, however, as it suggests that females may not need to invest in more mate assessment after mating, in particular if mate choice is not adaptive. We have added a statement to make this point here (line 432).

L367: a possible place for sperm quality as well?

We moved discussion on sperm quality to the discussion paragraph with differences between experienced and non-experienced females and the discussion about difference in traits between the ARTs.

---

## [Decision Letter · Decision Letter 1]

13 Jun 2023

Genotype and growth rate influence female mate preference in Xiphophorus multilineatus: potential selection to optimize mortality-growth rate tradeoff

PONE-D-23-01131R1

Dear Fitschen-Brown,

Thank you for this thorough revision, which has now been reexamined by myself and one of the reviewers. We both found the manuscript to be be suitable for publication. We’re pleased to inform you that your manuscript has been judged scientifically suitable for publication and will be formally accepted for publication once it meets all outstanding technical requirements.

Kind regards,

Quenton M. Tuckett

Academic Editor

PLOS ONE

---

## [Editor Report · Acceptance letter]

22 Jun 2023

PONE-D-23-01131R1 

Genotype and growth rate influence female mate preference in *Xiphophorus multilineatus*: potential selection to optimize mortality-growth rate tradeoff 

Dear Dr. Fitschen-Brown:

I'm pleased to inform you that your manuscript has been deemed suitable for publication in PLOS ONE. Congratulations! Your manuscript is now with our production department. 

Kind regards, 

on behalf of

Dr. Quenton M. Tuckett 

Academic Editor

PLOS ONE